# The impact of VUS reclassification on reproductive decision making

Alexandra Peyser[1]*, Kenan Onel[2], Avner Hershlag[3]

**1** Northwell, New Hyde Park, New York; Division of Reproductive Endocrinology and Infertility, Department of Obstetrics and Gynecology, North Shore University Hospital, Manhasset, New York; Zucker School of Medicine, Uniondale, New York, United States of America, **2** Roswell Park Comprehensive Cancer Center, Buffalo, New York, United States of America, **3** Division of Reproductive Endocrinology, Department of Obstetrics and Gynecology, Renaissance School of Medicine, Stony Brook University Hospital, Stony Brook, New York, United States of America

* apeyser@northwell.edu

## Abstract

### Introduction

Laboratories will occasionally reclassify a VUS to pathogenic (P) or likely pathogenic (LP), making it clinically actionable. Here, we aim to characterize the frequency of this reclassification in genes tested routinely during preconception counseling in order to help guide reproductive decision making.

### Design

Utilizing the American College of Medical Genetics (ACMG) 113-gene pre-conception panel, we conducted data analysis from ClinVar Miner (https://clinvarminer.genetics.utah.edu/). The numbers of VUS's reported for each gene were recorded over a 3-year period (2019, 2021, & 2022). In addition, data on the number of VUS's *in conflict* (VUS/P and VUS/LP) were recorded. The 10 genes with the most VUS's and the genes with the most frequent reporting discordance were compared over the 3 years,

### Results

There was a 103% increase in the number of VUS's reported (2019: 13,278, 2021: 22,434, 2022: 26,965) and a 235% increase in the number of VUS's in conflict (2019:387, 2021: 946, 2022:1297). The overall percent conflict increased significantly (2019: 2.9% vs. 2022: 4.8%). Nine genes among the top 10 with the most frequent VUS's remained the same over the 3-year period, while five out of the ten most frequent genes with VUS's in conflict remained the same.

**Data availability statement:** https://clinvarminer.genetics.utah.edu/ Within clinvarminer, searching by specific gene will display the number of VUS, pathogenic and likely pathogenic variants for each gene for each year of interest can be found.

**Funding:** The author(s) received no specific funding for this work.

**Competing interests:** The authors have declared that no competing interests exist.

## Conclusions

The rate of conflicting reporting of a VUS has increased in 3 years and is currently at 4.8%. A potential upgrade of a VUS to pathogenic or likely pathogenic may turn the variant "actionable," justifying testing embryos for the variant through PGT-M.

## Introduction

Due to an increase in preconception genetic testing, laboratories are now detecting novel sequence variants for an increasing number of genes associated with genetic disorders. The significance of these variants falls along a gradient indicating the likelihood that the variant is associated with a disease. In 2015, the American College of Medical Genetics (ACMG) published guidelines to standardize the classification of variants indicating the likelihood of being associated with disease ranging from pathogenic (P), likely pathogenic (LP), variant of uncertain significance (VUS), likely benign (LB) to benign (B) [1]. While variants classified as B/LB are not actionable and variants classified as P/LP are actionable, the finding of a VUS poses a dilemma, as its association with disease is unclear and may change over time as more individuals are sequenced. Patients with a family history of a heritable disease with a VUS in a gene associated with that disorder present a unique challenge to the reproductive geneticist: While embryos can be tested for any genetic variant through preimplantation genetic testing (PGT), VUS's are considered not actionable. Indeed, most PGT labs do not test embryos for a VUS.

Variants may be classified as VUS for several reasons: Either the effect of the specific genetic alteration on gene function is unknown, or there is insufficient genetic data to definitively confirm the pathogenicity of the variant [1]. VUS reporting during the pre-conception period presents a significant challenge to reproductive specialists. This holds true when both partners are carriers for a VUS or if one partners screen positive for an X-linked condition where male offspring will be affected. When a VUS is reported, the question arises whether pre-implantation genetic testing for monogenic disease (PGT-M) of embryos should be recommended to prevent the offspring from inheriting the variant. Since assisted reproductive technology (ART) has the unique capacity to identify genetic variants, in the embryo preconceptionally thus preventing their transmission to the offspring, the common reporting of VUS presents a difficult dilemma, especially due to the lack of specific guidelines by the professional societies [2,3]. Additionally, as data continues to accumulate, laboratories will occasionally reclassify a VUS to P or LP, both of which are considered clinically actionable, allowing the utilization of PGT-M to prevent vertical transmission to the offspring. The ongoing reclassification of VUS, coupled with conflicting reporting by testing laboratories of the pathogenic significance of variants has led to confusion regarding the need for preconception testing of embryos for the variant in question [4]. Since the use of wide carrier screening (CS) is becoming a staple of preconception screening, especially in the field of Assisted Reproductive Technologies (ART), where a couple will be screened for hundreds of disease causing variants, the

incidence VUS reporting keeps growing. Providers are facing an increasing need to explain the significance of VUSs and their actionability. There is a dilemma; what should the provider do when the variant is reported as VUS by one or more labs and as P or LP by another lab(s). Should conflicting reports present an opportunity to perform PGT-M? The objective of this study was to explore the extent of VUS reporting over a 3-year period for the recommended pre-conception genetic screening panel, and to specifically determine how many VUS's are in conflict between laboratories, making them potentially actionable. It is important to note that in this study, we analyzed VUS in conflict that were classified as a VUS by one laboratory and P or LP with another. We did not look at those that were in conflict in being classified as benign or likely benign because they were clinically irrelevant.

## Materials and methods

Utilizing the ACMG recommended 113 gene pre-conception panel [5], consisting of 97 autosomal recessive and 16 X-linked genetic conditions, we conducted data analysis from ClinVar Miner (https://clinvarminer.genetics.utah.edu), a web-based platform utilizing data from the National Center for Biotechnology Information's ClinVar archive (https://www.ncbi.nlm.nig.gov/clinvar) in December 2022. ClinVar, the main community-based repository of genomic knowledge is a shared variant interpretation database that is updated weekly with several thousand modifications of variant classifications.

The number of VUS's reported for each gene was recorded over a 3-year period (2019, 2021, and 2022) by one author (AP). Data for 2020 was not reported on ClinVar Miner (due to the COVID pandemic). In addition, data on the number of VUS's *in conflict* (defined as those reported as VUS by one submitter and as P or LP by another) were obtained. The top 10 genes with the highest number of VUS's per gene as well as the highest percent in conflict (defined as a VUS in conflict divided by the total number of VUS) were compared over the 3 years.

## Results

Over the study period, there was a 103% increase in the number of VUS's reported from the genes within the ACMG recommended panel (2019: 13,278, 2021: 22,434, 2022: 26,965) and a 235% increase in the number of VUS's in conflict (2019:387, 2021: 946, 2022: 1297) (Fig 1). The total percentage of VUS's in conflict increased over time from 2.9% in 2019 to 4.8% in 2022. Of the 113 genes on the panel, there was an increase over time in the number of genes with conflicting reporting (2019: 64, 2021:90, 2022:101) (Fig 2). The top 9 genes with the highest number of VUS's remained

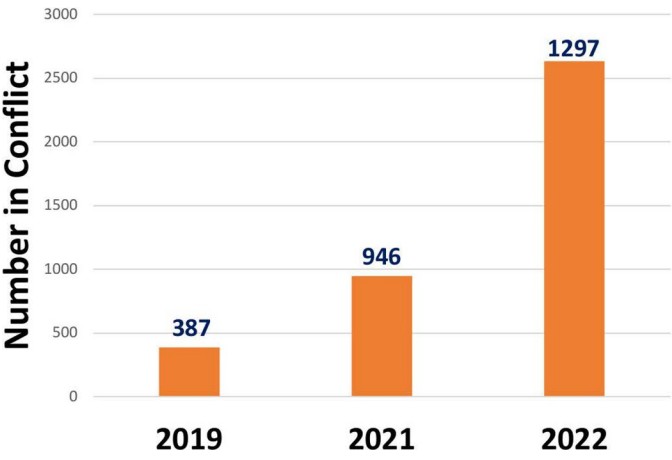

**Fig 1. Total VUS's in Conflict by Year.**

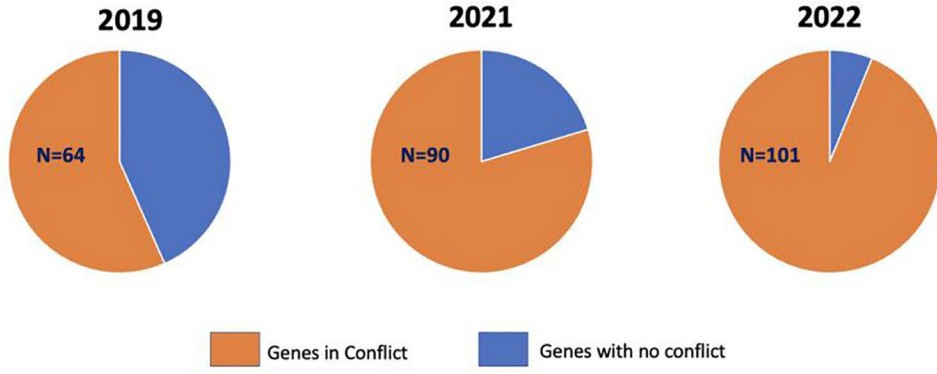

**Fig 2. Total Genes in Conflict by Year.**

the same over the 3 years (NEB, DMD, USH2A, BLM, PKHD1, CFTR, POLG, GAA, PCDH15) while the 10th gene differed each year (Fig 3). Of the top 10 genes with the highest percent in conflict, 5 remained in the top 10 over the 3-year period (Fig 4).

## Discussion

Over the study period, the number of VUS's reported for the ACMG-recommended preconception panel has increased significantly. In addition, the rate of conflicting reporting (VUS/P or VUS/LP) has increased by 235% and is currently at 4.8%. There exists no consensus on how reproductive medicine providers should counsel patients regarding VUS results on CS panels [6–8]. While an ever-expanding carrier-screening has become an integral part of preconception testing, we have witnessed an exponential increase in VUS reporting falling into an interpretive vacuum, given the absence of a broad consensus regarding the clinical management to VUS's and how should patients be counseled. Some argue that a VUS should be ignored during clinical decision making due to misleading consequences [9], while others have called for a more widely accepted consensus regarding management [10].

Currently, there is a lack of guidelines in the United States for determining whether PGT-M should be performed for a VUS. This has resulted in wide discrepancies between PGT-M laboratories as to whether or not they agree to perform PGT-M for a VUS [2]. Porto *et al.* interviewed nine genetic counselors from 5 different PGT-M laboratories regarding utilization of PGT-M for VUS [2]. The authors concluded that there was a significant variation in the eligibility policy regarding VUS for PGT-M between laboratories, and that there is a perceived lack of specific guidelines by the professional societies regarding counseling patients [2].

The American Society for Reproductive Medicine (ASRM) ethics committee opinion states that PGT-M is ethically justifiable in cases of serious conditions with no known interventions, as well as for cases of less serious or lower penetrance disorders due to reproductive liberty [11]. Additionally, the American College of Obstetricians and Gynecologists (ACOG) Committee Opinion from March 2020 states that PGT-M has clinical utility but did not make specific recommendations as to which variants should be tested for via PGT-M [12]. A recent ASRM committee opinion stated that offering PGT-M for VUS depends on multiple factors including how the VUS was identified, supporting classification evidence, familial penetrance, recurrence risk and supporting documentation. The document does not mention what to do when a VUS is labeled pathogenic one place and a VUS elsewhere [13].

At a single IVF center experience in Israel, data on couples presenting for PGT-M for a VUS was collected and analyzed [14]. From 2014–2019, a total of 45 couples requested PGT-M, of which 24 (52%) had a VUS. Twelve (50%) of the latter couples had an isolated VUS and 12 (50%) had a VUS diagnosed in addition to a disease-causing variant. Nine couples underwent PGT-M for an isolated VUS, 5 couples performed PGT-M for both pathogenic findings and a VUS.[12]

| Genes with highest # of VUS | | | | | |
|---|---|---|---|---|---|
| **2019** | | **2021** | | **2022** | |
| DMD | 871 | NEB | 1320 | NEB | 1567 |
| NEB | 707 | DMD | 1294 | DMD | 1553 |
| PKHD1 | 701 | USH2A | 1159 | USH2A | 1511 |
| USH2A | 580 | BLM | 981 | BLM | 1190 |
| CFTR | 578 | PKHD1 | 939 | PKHD1 | 1042 |
| PCDH15 | 492 | CFTR | 818 | CFTR | 958 |
| BLM | 488 | GAA | 638 | POLG | 763 |
| POLG | 426 | POLG | 620 | GAA | 744 |
| GAA | 411 | PCDH15 | 591 | PCDH15 | 712 |
| DYNC2H1 | 247 | ATP7B | 460 | CEP290 | 560 |

**Fig 3. Genes with the highest number of VUS by year.** The top 9 genes with the highest number of VUS's remained the same over the 3 years while the 10th gene (red) differed each year.

| Genes with Highest % Conflict | | | | | |
|---|---|---|---|---|---|
| **2019** | | **2021** | | **2022** | |
| PLP1 | 100 | PLP1 | 100 | **PAH** | 25.98 |
| **BTD** | **45.63** | **BTD** | **46.72** | **BTD** | **23.84** |
| **CYP27B1** | 36.84 | **CYP27B1** | 35.9 | RS1 | 22.22 |
| GALT | 26.92 | GALT | 25.86 | ASPA | 20 |
| FXN | 16.67 | RS1 | 25 | OCA2 | 16.9 |
| **PAH** | 14.42 | **PAH** | 22.13 | **ACADM** | 15.79 |
| GBA | 11.11 | ASPA | 20 | **CYP27B1** | 15.56 |
| **ACADM** | 11.02 | **ACADM** | 14.55 | **GJB2** | 14.04 |
| **GJB2** | 10.19 | **GJB2** | 13.82 | ACADVL | 13.95 |
| ASL | 9.09 | ACADVL | 12.05 | CFTR | 13.88 |

**Fig 4. Genes with the highest percent in conflict by year.** Of the top 10 genes with the highest percent in conflict, the 5 in bold remained in the top 10 over the 3-year period.

Another study which looked at 970 prenatal microarrays following invasive diagnostic prenatal testing found a VUS in 55 cases (5.8%) [15]. A VUS result creates challenges for providers to educate and counsel their patients appropriately as well as presenting difficult decisions for parents regarding how to proceed following a VUS result.

Whether VUS results should be disclosed in CS reporting remains an ongoing debate, especially within the prenatal context. VUS disclosure has the potential to increase parental distress as well create difficulties for genetic counselors as to how to advise patients [16]. In fact, reporting of VUS varies globally; the U.K and Belgium withhold VUS reporting whereas the Netherlands and the U.S. routinely report them [16]. Following a diagnosis of a VUS prenatally, patients have expressed concern about the child's health and development [17]. Furthermore, it has been reported that a substantial proportion of physicians lack a true understanding of the implications of a VUS result and do not feel comfortable disclosing a VUS to a patient [18]. Prior studies have shown that patients often misinterpret VUS results depending on how counseling was performed [19]. Physicians should be educated regarding VUS. Our disease prevention mission is handicapped by the currently inefficient setting. The development of clear guidelines by the professional societies will fill in an important void, enhance communication with patients and dramatically reduce frustration of patients, genetic counselors, and physicians.

Another dilemma reproductive specialists face is what to do when reclassification of a VUS occurs. Studies have reported the median time for reclassification is 39 months [17]. A previous study looked into reclassification data over 3.5 years (2016–2019) and found that 0.95% of VUS's were reclassified, of them 6% to P and 18% to LP, making these

variants clinically actionable, thus enabling the utilization of PGT-M [20]. Many studies have emphasized the clinical impact of reclassification on cancer genes [21–23]. In cases where embryos were frozen, often times following a biopsy for aneuploidy, a delayed reclassification to P or LP would raise the option of embryo thaw and re-biopsy. What happens when there is a discrepancy of the variant classification between laboratories; When one laboratory reports the variant as P or LP and another calls it is a VUS. Is PGT-M warranted in these cases? If conflicting reports exist, who should decide whether a VUS should be actionable? A prior study looked at discordant variant classifications between clinicians and genetic testing laboratories within a single cardiovascular disease clinic and found that 18% of variants were reported differently by the two groups, with 83% affecting the clinical care of the patient [24]. Our study demonstrates that currently, 4.8% of VUS's from the ACMG prenatal panel are in conflict between genetic testing laboratories, making them clinically actionable.

There is no formal reclassification process published yet by the ACMG or any other professional society. There is a missing link between the lab finding a variant and the clinical background. There is no mechanism set in place, nor a central registry where the familial inheritance of the disease in question is reported. Therefore, the determination of pathogenicity seems at this point to be haphazard in the lack of an organized database. A recent case series discussed 3 pediatric cases of adrenoleukodystrophy in patients conceived via the same oocyte donor, all found to be heterozygous of the same VUS which the donor was not originally screened for [24]. After the death of one child at the age of 5, the two other families from the same donor were notified by their clinic and the children were treated. This case highlights the gap in screening recommendations of gamete donors as well as the lack of standardized protocol for sharing any subsequent reclassification. When new evidence suggests pathogenicity of a VUS, the ACMG guidelines can be used to change the classification. Furthermore, reclassification of a VUS is highly dependent on testing laboratories. Some laboratories have an active variant reclassification process in which periodic updates to VUS classified variants is performed and if reclassification occurs, all providers on record for those affected are informed [25]. Whereas other laboratories use a passive reclassification system where the providers are the ones to inform the lab of any new information with regard to the variant [23]. This variation in reclassification process warrants a uniform system. It is incumbent upon providers to review the literature and rationales presented by their testing laboratory for classification interpretation and reinterpretation.

We may assume that given the exponential increase in VUS and VUS in conflict reporting these numbers are only going to increase several-fold within 5,10 and 20 years [26,3]. Importantly, since the rate of CS is increasing as well as the increase availability (and cost containment) of whole exome and whole genome sequencing is upon us, the absence of clear policy and guidelines is bound to lead to total chaos in the interpretation and clinical utilization of genetic data. We suggest a combined forum of ACMG and ASRM to propose a specific protocol for examining the pathogenicity of VUS's and a well-defined mechanism to determine, in a timely fashion, which VUS's should be upgraded.

The strengths of our study include that it is the first to describe the predicament of VUS and VUS in conflict within the field of reproductive medicine and more importantly, within the preconception period. We highlight the need for uniformity in the variant reclassification system in reproductive medicine. Limitations include the ever-changing variant classification and therefore our results may differ by the time this study is completed. In addition, our study was limited to the 3-year time period studied, further hampered by the lack of 2020 data due to the COVID pandemic. Furthermore, prior studies using ClinVar have revealed that variant types, penetrance, and a widely varied testing technology between genetic labs could affect the concordance rates between submitters, which could affect our results [27]. We did not report on those VUS's in conflict with a benign and likely benign classification as our focus was on those VUS that are actionable leading to potential use of PGT-M.

## Conclusion

The reporting of a VUS, especially with significant family history, presents a difficult dilemma to the reproductive physician: should action be taken to prevent inheritance to the offspring through PGT-M. The ACMG has not established a clear reclassification process for VUS's when conflicting reporting is documented on ClinVar. Therefore, a proactive approach to

a VUS may reveal that it has been reported as pathogenic elsewhere, making intervention through PGT-M a reasonable action, without having to wait for an official upgrade. Prior to each subsequent pregnancy, it is important to reevaluate previously reported VUS's, since they may have been reclassified.

> ### Key Message
>
> The rate of conflicting reporting of VUS on the pre-conception panel is increasing. A proactive approach to a VUS may reveal it is pathogenic elsewhere, making PGT-M a reasonable action. Prior to each subsequent pregnancy, it is important to reevaluate previously reported VUS's, since they may have been reclassified.

## Supporting information

**S1 Table. ClinVar_Data_ASRM22_REAL.**
(XLSX)

## Author contributions

**Conceptualization:** Alexandra Peyser, Kenan Onel, Avner Hershlag.

**Data curation:** Alexandra Peyser, Kenan Onel, Avner Hershlag.

**Formal analysis:** Alexandra Peyser, Kenan Onel, Avner Hershlag.

**Investigation:** Alexandra Peyser.

**Methodology:** Alexandra Peyser, Avner Hershlag.

**Project administration:** Alexandra Peyser, Avner Hershlag.

**Resources:** Alexandra Peyser.

**Supervision:** Alexandra Peyser, Kenan Onel, Avner Hershlag.

**Validation:** Alexandra Peyser, Avner Hershlag.

**Visualization:** Alexandra Peyser, Avner Hershlag.

**Writing – original draft:** Alexandra Peyser, Kenan Onel, Avner Hershlag.

**Writing – review & editing:** Alexandra Peyser, Avner Hershlag.

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
