## [Decision Letter · Decision Letter 0]

9 Sep 2024

PONE-D-24-31357The Impact of VUS Reclassification on Reproductive Decision MakingPLOS ONE

Dear Dr. Peyser,

Thank you for submitting your manuscript to PLOS ONE. After careful consideration, we feel that it has merit but does not fully meet PLOS ONE’s publication criteria as it currently stands. Therefore, we invite you to submit a revised version of the manuscript that addresses the points raised during the review process.

We look forward to receiving your revised manuscript.

Kind regards,

Nejat Mahdieh

Academic Editor

PLOS ONE

Journal Requirements:

“MINING” VUS’S FOR PATHOGENICITY: CAN INTER-LAB CONFLICTS RENDER VARIANTS OF UNCERTAIN SIGNIFICANCE (VUS’S) PATHOGENIC AND THEREFORE, ACTIONABLE? - https://doi.org/10.1016/j.fertnstert.2022.08.168

(among others)

In your revision ensure you cite all your sources (including your own works), and quote or rephrase any duplicated text outside the methods section. Further consideration is dependent on these concerns being addressed.

Reviewers' comments:

Reviewer's Responses to Questions

**Comments to the Author**

1. Is the manuscript technically sound, and do the data support the conclusions?

Reviewer #1: Partly

Reviewer #2: Yes

2. Has the statistical analysis been performed appropriately and rigorously? 

Reviewer #1: N/A

Reviewer #2: Yes

3. Have the authors made all data underlying the findings in their manuscript fully available?

Reviewer #1: Yes

Reviewer #2: Yes

4. Is the manuscript presented in an intelligible fashion and written in standard English?

Reviewer #1: Yes

Reviewer #2: Yes

5. Review Comments to the Author

Reviewer #1: The authors of the article entitled: The Impact of VUS Reclassification on Reproductive Decision Making” gives a view of the 3 years of VUS reports and report the increase of VUS in reproductive genes. “When a VUS is reported, the question arises whether pre-implantation genetic testing for monogenic disease (PGT-M) of embryos should be recommended to prevent the offspring from inheriting the variant.” As mentioned the variants with Pathogenic and LP effect are actionable.

Please clarify how many VUS variants of the genes were changed during these years to P/LP. Note that these are actionable for PGT.

Reviewer #2: Variant discovery resulting in VUS during preconception recessive and X-linked carrier testing is an important issue in the medical literature. Defining precisely what the authors consider to be carrier screening would benefit this paper as there are a few areas in which the definitions cross boundaries into other types of genetic testing in the prenatal setting. Prenatal genetic testing differs across jurisdictions – the paper would benefit if the authors state what specific issue within their jurisdiction needs to be resolved. Please find some comments on the proposed paper:

Abstract

“Laboratories will occasionally upgrade a VUS to pathogenic (P) or likely pathogenic (LP), making it clinically actionable.”

- Variant reclassification may result in the changed classification of any variant to any other distinct category and is not restricted to an “upgrade”

Introduction

There are three distinct entities within genetic testing in the prenatal setting that seem to be viewed interchangeably by the authors. The aim of the authors study appears to be point 1 below, and this study conflates an overlap between 2 & 3:

1. Asymptomatic recessive and X-linked carrier testing within prenatal medicine which is indicated for an unaffected (asymptomatic) individual without any unprobed family history of a genetic condition

2. Non-invasive prenatal testing using chromosomal microarray technology

3. VUS that are uncovered during the course of diagnostic testing in an affected individual, thereby leaving the case unsolved and no option for prenatal/preconception genetic testing for a known condition available to closely related family members. (Clinically, risk could be established for an unaffected but closely related couple using carrier frequencies to estimate risks of a rare recessive condition).

e.g. “Patients with a family history of a heritable disease with a VUS in a gene associated with that disorder present a unique challenge to the reproductive geneticist”.

- The proper steps for a genetic diagnosis in a family for which preconception genetic testing can be offered is to make a diagnosis in the affected individual. The focus of the paper appears to be on asymptomatic recessive and X-linked carrier testing in prenatal medicine which is performed independent of a diagnosis in consideration. Therefore this statement appears to confuse diagnostic versus carrier testing. (Screening for autosomal recessive and X-linked conditions during pregnancy and preconception: a practice resource of the American College of Medical Genetics and Genomics (ACMG) )

“VUS’s are generally considered not actionable. Indeed, most PGT labs do not test embryos for a VUS” –

- VUS are always considered not actionable at the point of classification. VUS testing of an embryo may occur in select instances and is not really generalisable to the point of this paper as a whole (Laboratory testing for preconception/prenatal carrier screening: A technical standard of the American College of Medical Genetics and Genomics (ACMG))

“VUS reporting during the pre-conception period presents a significant challenge to reproductive specialists. When a VUS is reported, the question arises whether pre-implantation genetic testing for monogenic disease (PGT-M) of embryos should be recommended to prevent the offspring from inheriting the variant.”

- The authors should emphasise that risk for the genetic variants classified as VUS during prenatal carrier testing is only significant in light of their partner’s status. Therefore, this is a leap to say that a VUS in a recessive gene may have clinical implications when you don’t consider the partner’s risk.

“the common reporting of VUS presents a difficult dilemma”

- Do the authors have a reference for how commonly VUS are reported in the prenatal setting?

“The ongoing reclassification of VUS, coupled with conflicting reporting by testing laboratories of the pathogenic significance of variants has led to confusion regarding the need for preconception testing of embryos for the variant in question (3).”

- The authors repeatedly reference laboratory reporting of discordant classification of the same variant. This is a separate issue to VUS reclassification and although it is examined in the methods of this study, may not contribute to ultimate genetic counselling. Can the authors please define the significance of variant discordance within the context of the study aims?

- “We did not report on those VUS’s in conflict with a benign and likely benign classification as our focus was on those VUS that are actionable leading to potential use of PGT-M.” – I note this as the final statement of the Discussion which provides useful information about the aim of the paper overall and would be important in the Introduction to clarify the aims.

“There is a dual dilemma; first, how should the patient be counselled when the variant is reported as VUS by one or more labs and as P or LP by another lab(s)”

- Genetic counselling is clinically confined to the genetic test report and the consideration of the testing centre and clinician interpretation.

“The second dilemma is how to relate to a VUS reported, for instance, for a cancer gene, where there is a significant familial history of the cancer associated with the gene (e.g., breast/ovarian cancer and BRCA1/2; colon cancer and APC).”

- This would require clarification as the authors previously state that the ACMG recommended gene panel for preconception genetic testing is the foundation of this study. However, cancer associated genes are not included in this panel. Can the authors please clarify this?

Materials and methods:

“The top 10 genes with the highest number of VUS’s per gene as well as the highest percent in conflict (defined as a VUS in conflict divided by the total number of VUS) were compared over the 3 years.”

- Why did the authors restrict their analysis to 10 genes?

- Why did the authors confine the time period to 3 years of data?

- Discordant submissions in ClinVar are not the same as VUS classification of variants and the authors should note the distinction

- the methods could be improved by comprehensively examining VUS detected for genes within the recommended gene panel

Discussion:

“VUS disclosure has the potential to increase parental distress as well create difficulties for genetic counselors as to how to advise patients (15). I”

- This distress seems to arise from the disclosure of CNV VUS detected during NIPT for an active pregnancy. Do the authors feel that this may extend to preconception counselling for what may be non clinically relevant VUS in recessive genes?

“. Our disease (cancer included) prevention mission is handicapped by the currently inefficient setting”

- Again, can the authors please clarify what preconception carrier testing methods report pathogenic or likely pathogenic variants in cancer related genes?

“Reporting to ClinVar is not mandatory”.

- However ClinVar still represents a rich repository of human genetic variation with over 4 million genetic variants submitted. It has proven use in the literature as an adjunct to variant classification and therefore does not represent a limitation of variant classification or reclassification.

“We may assume that given the exponential increase in VUS and VUS-in-conflict reporting, these numbers are only going to increase several-fold within 5,10 and 20 years.”

- Standardised guidelines and improved sequencing and bioinformatic techniques are resolving unsolved variation within the human genome. Do the authors have a reference to support this statement? Otherwise, it is in conflict with the current published literature.

“What will happen if VUS’s in conflict reach 25% of all cases?”

- Can the authors clarify the implications of this statement? What would happen? And is it likely to happen?

“Currently, many genetic labs do not report all VUS’s and therefore, the true incidence of potentially upgradable VUS’s is difficult to determine but likely, involves many more under-reported cases.”

- The study methods use ClinVar miner which is taken from ClinVar. ClinVar is a variant submission database, not a genetic test report database. Therefore, submitting labs and facilities submit any variant uncovered during genetic testing, and not just those that are clinically relevant for the test performed. Therefore this statement is inaccurate as VUS within ClinVar are more likely to closely represent all VUS detected during genetic diagnostic testing and research settings and not just those that reach the eyes of the clinician and patient on the final genetic test report.

6. PLOS authors have the option to publish the peer review history of their article (what does this mean? ). If published, this will include your full peer review and any attached files.

**Do you want your identity to be public for this peer review?** For information about this choice, including consent withdrawal, please see our Privacy Policy .

Reviewer #1: No

Reviewer #2: No

---

## [Author Response · Author response to Decision Letter 1]

14 Feb 2025

Thank you for your consideration of our manuscript. We have revised the manuscript to accommodate your suggestions and greatly appreciate each review. Specific responses to comments are itemized below.

Reviewer #1: The authors of the article entitled: The Impact of VUS Reclassification on Reproductive Decision Making” gives a view of the 3 years of VUS reports and report the increase of VUS in reproductive genes. “When a VUS is reported, the question arises whether pre-implantation genetic testing for monogenic disease (PGT-M) of embryos should be recommended to prevent the offspring from inheriting the variant.” As mentioned the variants with Pathogenic and LP effect are actionable.

Please clarify how many VUS variants of the genes were changed during these years to P/LP. Note that these are actionable for PGT.

Thank you for this comment. The main focus of our study was reporting on VUS in conflict (VUS/P and VUS/LP) that are potentially actionable. We did not look specifically at how many of these gene were changed to P/LP. However a prior study did study this and it is quoted in our discussion on page 9: “A previous study looked into reclassification data over 3.5 years (2016-2019) and found that 0.95% of VUS’s were reclassified, of them 6% to P and 18% to LP, making these variants clinically actionable, thus enabling the utilization of PGT-M (19).”

Reviewer #2: Variant discovery resulting in VUS during preconception recessive and X-linked carrier testing is an important issue in the medical literature. Defining precisely what the authors consider to be carrier screening would benefit this paper as there are a few areas in which the definitions cross boundaries into other types of genetic testing in the prenatal setting. Prenatal genetic testing differs across jurisdictions – the paper would benefit if the authors state what specific issue within their jurisdiction needs to be resolved. Please find some comments on the proposed paper:

Thank you. We have added a sentence to clarify that our carrier screening is performed for patients attempting a pregnancy. “Since the use of wide carrier screening (CS) is becoming a staple of preconception screening, especially in the field of Assisted Reproductive Technologies (ART), where a couple will be screened for hundreds of disease causing variants, the incidence of VUS reporting keeps growing.”

Abstract

“Laboratories will occasionally upgrade a VUS to pathogenic (P) or likely pathogenic (LP), making it clinically actionable.”

- Variant reclassification may result in the changed classification of any variant to any other distinct category and is not restricted to an “upgrade”

Thank you for this comment. We have changed the wording “upgrade” to “reclassify.”

Introduction

There are three distinct entities within genetic testing in the prenatal setting that seem to be viewed interchangeably by the authors. The aim of the authors study appears to be point 1 below, and this study conflates an overlap between 2 & 3:

1. Asymptomatic recessive and X-linked carrier testing within prenatal medicine which is indicated for an unaffected (asymptomatic) individual without any unprobed family history of a genetic condition

2. Non-invasive prenatal testing using chromosomal microarray technology

3. VUS that are uncovered during the course of diagnostic testing in an affected individual, thereby leaving the case unsolved and no option for prenatal/preconception genetic testing for a known condition available to closely related family members. (Clinically, risk could be established for an unaffected but closely related couple using carrier frequencies to estimate risks of a rare recessive condition).

e.g. “Patients with a family history of a heritable disease with a VUS in a gene associated with that disorder present a unique challenge to the reproductive geneticist”.

- The proper steps for a genetic diagnosis in a family for which preconception genetic testing can be offered is to make a diagnosis in the affected individual. The focus of the paper appears to be on asymptomatic recessive and X-linked carrier testing in prenatal medicine which is performed independent of a diagnosis in consideration. Therefore this statement appears to confuse diagnostic versus carrier testing. (Screening for autosomal recessive and X-linked conditions during pregnancy and preconception: a practice resource of the American College of Medical Genetics and Genomics (ACMG) )’

Yes, our main focus is point 1. However, often patients present with a family history of disease with a VUS detected and would like to do PGT-M. Therefore, we raise this point in the paper.

“VUS’s are generally considered not actionable. Indeed, most PGT labs do not test embryos for a VUS” –

- VUS are always considered not actionable at the point of classification. VUS testing of an embryo may occur in select instances and is not really generalisable to the point of this paper as a whole (Laboratory testing for preconception/prenatal carrier screening: A technical standard of the American College of Medical Genetics and Genomics (ACMG))

We have removed the word “generally.”

“VUS reporting during the pre-conception period presents a significant challenge to reproductive specialists. When a VUS is reported, the question arises whether pre-implantation genetic testing for monogenic disease (PGT-M) of embryos should be recommended to prevent the offspring from inheriting the variant.”

- The authors should emphasise that risk for the genetic variants classified as VUS during prenatal carrier testing is only significant in light of their partner’s status. Therefore, this is a leap to say that a VUS in a recessive gene may have clinical implications when you don’t consider the partner’s risk.

We have added an additional statement following that sentence explaining that this risk to the offspring is dependent on both a carrier status of both male and female partners. The exception is with X-linked genes where only male offspring are affected regardless.

“the common reporting of VUS presents a difficult dilemma”

- Do the authors have a reference for how commonly VUS are reported in the prenatal setting?

We have cited a reference that states that 41% of patients had a VUS reported.

“The ongoing reclassification of VUS, coupled with conflicting reporting by testing laboratories of the pathogenic significance of variants has led to confusion regarding the need for preconception testing of embryos for the variant in question (3).”

- The authors repeatedly reference laboratory reporting of discordant classification of the same variant. This is a separate issue to VUS reclassification and although it is examined in the methods of this study, may not contribute to ultimate genetic counselling. Can the authors please define the significance of variant discordance within the context of the study aims.

Later on in the introduction, we state “how should the patient be counselled when the variant is reported as VUS by one or more labs and as P or LP by another lab(s)… Should conflicting reports present an opportunity to perform PGT-M?” This is the ultimate conflict that reproductive endocrinologist often have.

- “We did not report on those VUS’s in conflict with a benign and likely benign classification as our focus was on those VUS that are actionable leading to potential use of PGT-M.” – I note this as the final statement of the Discussion which provides useful information about the aim of the paper overall and would be important in the Introduction to clarify the aims.

Thank you for this comment. We have added the statement to further clarify this in the last sentence in the introduction.

“There is a dual dilemma; first, how should the patient be counselled when the variant is reported as VUS by one or more labs and as P or LP by another lab(s)”

- Genetic counselling is clinically confined to the genetic test report and the consideration of the testing centre and clinician interpretation.

We have changed this to what should the provider in this case, as opposed to how the patient should be counselled. Conflicting reports on a variant make genetic counseling difficult, since the provider cannot be certain as to the significant of the variant, therefore inevitably setting the stage for confusion and frustration on the receiving end that is the patient. In the lack of clear genetic authority determining whether the variant is pathogenic (LP or P), this confusion is unsettling.

“The second dilemma is how to relate to a VUS reported, for instance, for a cancer gene, where there is a significant familial history of the cancer associated with the gene (e.g., breast/ovarian cancer and BRCA1/2; colon cancer and APC).”

- This would require clarification as the authors previously state that the ACMG recommended gene panel for preconception genetic testing is the foundation of this study. However, cancer associated genes are not included in this panel. Can the authors please clarify this?

We have removed this statement, as our main focus was on the ACMG recommended panel.

Materials and methods:

“The top 10 genes with the highest number of VUS’s per gene as well as the highest percent in conflict (defined as a VUS in conflict divided by the total number of VUS) were compared over the 3 years.”

- Why did the authors restrict their analysis to 10 genes?

For the sake of high incidence reporting, we chose the genes with the most frequent VUS as we have done so in previous published reports on genetic screening.

- Why did the authors confine the time period to 3 years of data?

The 3 years of data was all the data that was available on Clinvar Miner at the time of collection. The year of the pandemic (2020), there was minimal data collection.

- Discordant submissions in ClinVar are not the same as VUS classification of variants and the authors should note the distinction

This is stated in the material and methods: “data on the number of VUS’s in conflict (defined as those reported as VUS by one submitter and as P or LP by another) were obtained.”

- the methods could be improved by comprehensively examining VUS detected for genes within the recommended gene panel

We did examine the VUS for the recommended 113 pre-conception gene panel.

Discussion:

“VUS disclosure has the potential to increase parental distress as well create difficulties for genetic counselors as to how to advise patients (15). I”

- This distress seems to arise from the disclosure of CNV VUS detected during NIPT for an active pregnancy. Do the authors feel that this may extend to preconception counselling for what may be non clinically relevant VUS in recessive genes?

Similar concerns exist with the presence of the unknown significance of a variant whether in the preconception as well as the prenatal periods. Patients planning on having a child through ART who are being told that a variant is not actionable and may actually be inherited by their offspring may feel distressed and concerned about the genetic health of their baby, especially if that variant is reported as pathogenic by a different lab than they were tested in. This distress can be inherent in them before and during the entire pregnancy.

“. Our disease (cancer included) prevention mission is handicapped by the currently inefficient setting”

- Again, can the authors please clarify what preconception carrier testing methods report pathogenic or likely pathogenic variants in cancer related genes?

We have removed this as we did not look at cancer genes.

“Reporting to ClinVar is not mandatory”.

- However ClinVar still represents a rich repository of human genetic variation with over 4 million genetic variants submitted. It has proven use in the literature as an adjunct to variant classification and therefore does not represent a limitation of variant classification or reclassification.

We have removed this comment.

“We may assume that given the exponential increase in VUS and VUS-in-conflict reporting, these numbers are only going to increase several-fold within 5,10 and 20 years.”

- Standardised guidelines and improved sequencing and bioinformatic techniques are resolving unsolved variation within the human genome. Do the authors have a reference to support this statement? Otherwise, it is in conflict with the current published literature.

Yes, we have references to support the statement and have added them. (References 26 and 27)

“What will happen if VUS’s in conflict reach 25% of all cases?”

- Can the authors clarify the implications of this statement? What would happen? And is it likely to happen?

We have removed this statement as well.

“Currently, many genetic labs do not report all VUS’s and therefore, the true incidence of potentially upgradable VUS’s is difficult to determine but likely, involves many more under-reported cases.”

- The study methods use ClinVar miner which is taken from ClinVar. ClinVar is a variant submission database, not a genetic test report database. Therefore, submitting labs and facilities submit any variant uncovered during genetic testing, and not just those that are clinically relevant for the test performed. Therefore this statement is inaccurate as VUS within ClinVar are more likely to closely represent all VUS detected during genetic diagnostic testing and research settings and not just those that reach the eyes of the clinician and patient on the final genetic test report.

We very much appreciate your insight into the ClinVar reporting. We have removed the statement. This raises a significant clinical and ethical question. First, should a patient be unaware of the existence of a VUS that is not reported to her or him in the lab results but is reported to Clin Var. Secondly, what if the unreported VUS that is sent to ClinVar but is blinded to the patient conflicts with the same variant that is reported as LP or P by another lab. This phenomenon puts both the provider as well as the patient at a disadvantage.

---

## [Decision Letter · Decision Letter 1]

10 Mar 2025

The Impact of VUS Reclassification on Reproductive Decision Making

PONE-D-24-31357R1

Dear Dr. Peyser,

We’re pleased to inform you that your manuscript has been judged scientifically suitable for publication and will be formally accepted for publication once it meets all outstanding technical requirements.

Kind regards,

Nejat Mahdieh

Academic Editor

PLOS ONE

Additional Editor Comments (optional):

Reviewers' comments:

Reviewer's Responses to Questions

**Comments to the Author**

1. If the authors have adequately addressed your comments raised in a previous round of review and you feel that this manuscript is now acceptable for publication, you may indicate that here to bypass the “Comments to the Author” section, enter your conflict of interest statement in the “Confidential to Editor” section, and submit your "Accept" recommendation.

Reviewer #3: All comments have been addressed

2. Is the manuscript technically sound, and do the data support the conclusions?

Reviewer #3: Yes

3. Has the statistical analysis been performed appropriately and rigorously? 

Reviewer #3: Yes

4. Have the authors made all data underlying the findings in their manuscript fully available?

Reviewer #3: Yes

5. Is the manuscript presented in an intelligible fashion and written in standard English?

Reviewer #3: Yes

6. Review Comments to the Author

Reviewer #3: (No Response)

7. PLOS authors have the option to publish the peer review history of their article (what does this mean? ). If published, this will include your full peer review and any attached files.

**Do you want your identity to be public for this peer review?** For information about this choice, including consent withdrawal, please see our Privacy Policy .

Reviewer #3: **Yes: ** Farnoosh Emami

---

## [Editor Report · Acceptance letter]

PONE-D-24-31357R1

PLOS ONE

Dear Dr. Peyser,

I'm pleased to inform you that your manuscript has been deemed suitable for publication in PLOS ONE. Congratulations! Your manuscript is now being handed over to our production team.

Kind regards,

on behalf of

Dr. Nejat Mahdieh

Academic Editor

PLOS ONE